# Host gastric corpus microenvironment facilitates *Ascaris suum* larval hatching and infection in a murine model

Yifan Wu[1], Grace Adeniyi-Ipadeola[2], Mahliyah Adkins-Threats[3,4,5], Matthew Seasock[6], Charlie Suarez-Reyes[1], Ricardo Fujiwara[7], Maria Elena Bottazzi[1,2,8], Lizhen Song[6], Jason C. Mills[3,4,5], Jill E. Weatherhead [1,8,9] *

1 Department of Pediatrics, Division of Pediatric Tropical Medicine, Baylor College of Medicine, Houston, Texas, United States of America, 2 Department of Molecular Virology and Microbiology, Baylor College of Medicine, Houston, Texas, United States of America, 3 Department of Medicine, Section of Gastroenterology, Baylor College of Medicine, Houston, Texas, United States of America, 4 Departments of Pathology & Immunology, Baylor College of Medicine, Houston, Texas, United States of America, 5 Department of Molecular and Cellular Biology, Baylor College of Medicine, Houston, Texas, United States of America, 6 Department of Medicine, Immunology, Pathology, Baylor College of Medicine, Houston, Texas, United States of America, 7 Departamento de Parasitologia, Universidade Federal de Minas Gerais, Belo Horizonte, Brazil, 8 National School of Tropical Medicine, Baylor College of Medicine, Houston, Texas, United States of America, 9 Department of Medicine, Section of Infectious Diseases, Baylor College of Medicine, Houston, Texas, United States of America

* weatherh@bcm.edu

**Data Availability Statement:** All relevant data are within the paper and its supporting information files.

## Abstract

Ascariasis (roundworm) is the most common parasitic helminth infection globally and can lead to significant morbidity in children including chronic lung disease. Children become infected with *Ascaris* spp. via oral ingestion of eggs. It has long been assumed that *Ascaris* egg hatching and larval translocation across the gastrointestinal mucosa to initiate infection occurs in the small intestine. Here, we show that *A. suum* larvae hatched in the host stomach in a murine model. Larvae utilize acidic mammalian chitinase (AMCase; acid chitinase; *Chia*) from chief cells and acid pumped by parietal cells to emerge from eggs on the surface of gastric epithelium. Furthermore, antagonizing AMCase and gastric acid in the stomach decreases parasitic burden in the liver and lungs and attenuates lung disease. Given *Ascaris* eggs are chitin-coated, the gastric corpus would logically be the most likely organ for egg hatching, though this is the first study directly evincing the essential role of the host gastric corpus microenvironment. These findings point towards potential novel mechanisms for therapeutic targets to prevent ascariasis and identify a new biomedical significance of AMCase in mammals.

## Author summary

Ascariasis is the most common helminth infection and leads to significant global morbidity. Current therapies specifically target intraluminal adult intestinal worms and do not provide sufficient protection against the *Ascaris* larva. To reduce Ascaris-induced

**Funding:** This project was funded by US National Institutions of Health grants K08 AI143968-01 (to J.E.W.) and R01 DK105129 and P30 DK056338 (to J.C.M.). The funders had no role in study design, data collection and analysis, decision to publish, or preparation of the manuscript.

**Competing interests:** The authors have declared that no competing interests exist.

morbidity, novel interventions are needed that target Ascaris larval migration which calls for a better understanding of the *Ascaris* larval migration cycle. After oral ingestion of *Ascaris* eggs, previous studies have indicated that *Ascaris* larval migration initiates in the small intestine. However, our study demonstrates that *Ascaris* larva begin the migration cycle by hatching in the stomach by using the host's acidic mammalian chitinase and stomach acid to help degrade the chitinous egg shell. The larvae subsequently translocate across the stomach mucosa in this mouse model. Our study provides the first insight that the gastric corpus microenvironment plays a critical role in *Ascaris* egg hatching and initiation of infection, and will aid in the identification of novel drug targets.

## Introduction

Ascariasis (roundworm) is the most common parasitic helminth infection globally, affecting approximately 500 million people [1]. In endemic regions, people are infected in infancy and endure recurrent infection throughout childhood [2,3]. Children become infected with either *Ascaris lumbricoides*, the human roundworm, or *Ascaris suum*, the porcine roundworm, via oral ingestion of eggs that are ubiquitous within the environment [4,5]. Once ingested, *Ascaris* larvae hatch in the gastrointestinal tract and undergo a highly immunogenic, transient larval migratory phase through the host liver and host lungs before returning to the intestines and developing into adult worms [6]. High *Ascaris* worm burden and significant re-infection rates in children living in endemic regions leads to profound lifelong morbidity [7,8]. *Ascaris* larval migration through the lungs is associated with the development of several pulmonary diseases like asthma, while chronic intestinal ascariasis causes growth stunting, cognitive delays, malnutrition, abdominal pain and obstruction [3,6,9,10]. Together, these ascariasis-related pathologies lead to 800,000 disability adjusted life years (DALYS) worldwide [1].

Current anthelminthic therapy such as benzimidazoles used in mass drug administration campaigns specifically target intraluminal, adult intestinal worms. However, these anthelminthic therapies do not prevent *Ascaris* infection and do not provide sufficient protection against the *Ascaris* larval migration cycle because intestinal absorption of the drugs tend to be poor, and drug treatment has to be precisely timed to the relatively rapid transitions during larval stages [11]. As a result, children living in endemic regions are vulnerable to repeated bouts of *Ascaris* infection and *Ascaris*-induced morbidity [7,8]. Understanding the mechanisms that aid in *Ascaris* infectivity and the *Ascaris* larval migratory cycle could identify therapeutic targets to reduce *Ascaris* infection and *Ascaris*-induced morbidity in children around the world.

The molecular mechanisms of *Ascaris* egg hatching and translocation across the gastrointestinal mucosa remains largely unknown. Prior studies suggest that *Ascaris* larvae hatch in the small intestines or the colon and translocate across the intestinal mucosa to initiate the larval migration cycle [6,12,13]. However, a previous study demonstrated that repeated oral inoculation of piglets with *Ascaris* eggs can lead to the development of gastric ulcers, but the mechanism of ulcer development remains unknown [14]. In this study, we demonstrate that *A. suum* larvae utilize the gastric corpus microenviroment to aid in larval hatching from the ingested eggs and subsequently translocate across gastrointestinal mucosa using our established *A. suum* larval migration murine model.

## Results

### Initiation of *A. suum* larval migration occurs in the mouse stomach

After ingestion, *Ascaris* eggs hatch in the gastrointestinal tract, translocate across the mucosa and migrate to the liver via the portal vasculature [15]. A prior mouse study used Evan's blue dye (EBD) to demonstrate increased intestinal vasculature permeability following ingestion of *Ascaris* eggs to suggest *Ascaris* larval translocate across the intestines [12]. To confirm these findings, we utilized our standard *Ascaris* larval migration murine model in which BALB/c mice were challenged with 2,500 *A. suum* eggs via oral gavage [10,15]. We measured the permeability of the gastrointestinal tract vasculature of mice 1 day post infection by introducing EBD intravenously 1 hour prior to euthanasia and measuring total extravasated dye in homogenized tissue gastrointestinal segments at 620 nm by spectrophotometer after whole-body perfusion [12]. Surprisingly, in contrast to the previous study using the EBD technique [12], total dye in tissue was not increased in any segment of the intestines suggesting a lack of vascular permeability in the intestines following *Ascaris* infection (**Fig 1A**). However, we found increased EBD extravasation into the stomach tissue (**Fig 1B**). This result implied that the stomach may play a role in *Ascaris* larval hatching and mucosal translocation leading to infection.

To verify this hypothesis, using the *Ascaris* murine model, defined segments of the gastrointestinal tract from the esophagus to the colon were examined pathologically for evidence of *Ascaris* invasion over 4 days. We found no evidence of *Ascaris* larvae or mucosal damage in the esophagus or intestinal segments at any time point post infection 15 minutes, 30 minutes,

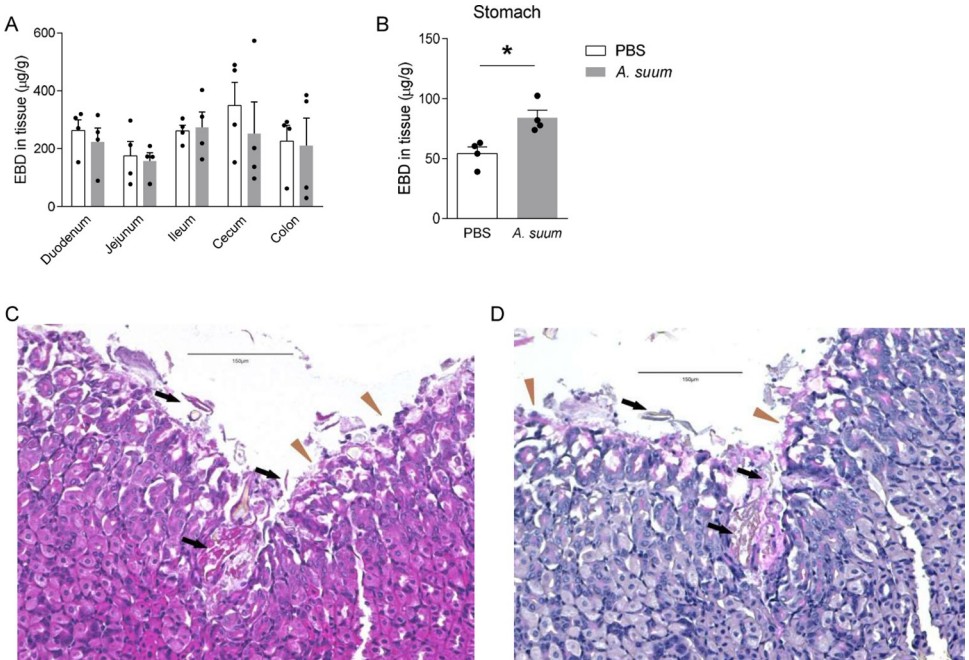

**Fig 1.** ***Ascaris suum* larva hatch in and migrate through the stomach: BALB/c mice were challenged by oral gavage with 2,500 eggs of *Ascaris* once. (A-B)** Evan's blue dye (EBD) was introduced intravenously to mice 1 day post infection. Recovery of EBD from **(A)** the intestine segments and **(B)** the stomach were quantified. **(C)** H&E and **(D)** PAS staining were performed on stomach sections 30 minutes post infection. Brown triangles indicate foveolar cells and black arrows indicate larval penetration across gastric mucosa. (n≥4, mean±S.E.M, *p<0.05 using two-tailed Student's t-test. Magnification: 200×. Scale bar: 150μm. Data are shown as representative of three independent experiments).

4 hours, 14 hours, 24 hours and 4 days (**S1 Fig–demonstrates histology 12 hours, 24 hours and 4 days post infection**). However, in the stomach, egg fragments and hatched *Ascaris* larvae aligned along the gastric corpus mucosa at 15 minutes post infection and larval structures were observed penetrating the gastric corpus mucosa as soon as 30 minutes post infection using H&E staining (**Fig 1C**). In foci where larvae were hatching and beginning to invade the gastric corpus epithelium, surface (foveolar) cells were damaged with increased PAS-positive mucin evident (**Fig 1D**). Together, these data suggest that *A. suum* eggs hatch in the mouse stomach, not intestines, with eggs abutting or adhering to the foveolar cells, and larvae quickly penetrate the gastric corpus mucosa to initiate larval migration.

## Gastric AMCase and gastric acid promote *A. suum* egg hatching

The gastric corpus mucosal barrier depends on repeated invaginated gastric units composed of mucous producing surface foveolar cells, stem cells, enteroendocrine cells, acid-secreting parietal cells, and pepsinogen and digestive-enzyme-secreting chief cells [16–18]. Eggs of *Ascaris spp*. are covered by a thick layer of chitin that ensures structural integrity in prolonged, harsh environmental conditions [19,20] but which must be degraded for *Ascaris* larvae to hatch and cause infection. To determine the mechanism of *Ascaris* egg hatching in the gastric corpus, we focused on the acidic mammalian chitinase (AMCase), a pH-dependent, protease-resistant, glycosidase, found in the lungs and the stomach, that hydrolyzes inhaled or ingested chitin [21–24]. Within the stomach AMCase is secreted only by chief cells which themselves reside only in the gastric corpus [22,25]. Because the gastric corpus is also the site of acid production, and AMCase functions optimally at low pH, we reasoned that the corpus would be the ideal anatomical location for chitin-coated *Ascaris* eggs to hatch. Chief cells secrete AMCase to hydrolyze ingested chitin-containing material, allowing for digestion of chitin-coated food sources (like insects and crustaceans) as well as offering potential protection against chitin-coated fungi[13,][24]. Gastric acid secretion is a highly regulated process maintained through a $H^+/K^+$ ATPase, a parietal cell proton pump [26]. Together, AMCase and gastric acid play an integral role in initiating digestion of chitinous food sources as well as composing the gastric barrier resistance against invasive chitinous pathogens [23,26,27]. Thus, we first evaluated the effect of AMCase in acidic conditions on *A. suum* egg hatching *in vitro*. We treated *A. suum* eggs with human AMCase (9859-GH-050, R&D systems, Minneapolis, MN) or control at pH = 2 or pH = 7 at 37°C overnight and measured larval hatching [24]. AMCase treatment at pH = 2 significantly increased the hatch rate of sham-treated eggs by almost 3-fold, which was significantly greater as compared to AMCase-treated eggs at neutral pH (**Fig 2A**). In addition to AMCase, chief cells also secrete the inactive zymogen pepsinogen, which is cleaved to pepsin in acidic gastric conditions and aids in protein digestion [24,26]. To evaluate the role of pepsin in larval hatching, we treated *A. suum* eggs with 4 or 8 μg/ml of pepsin at pH = 2 at 37°C overnight and measured larval hatching in vitro [24]. Unlike treatment with AMCase, treatment with pepsin did not aid in larval hatching (**S2 Fig**).

To determine if the immediate post-gastric intestinal microenvironment also enhanced egg hatching *in vitro*, we incubated *A. suum* eggs in PBS with a pH = 2 for 30 minutes. The acidic solution was removed and eggs were resuspended in neutral solution containing trypsin, bile salt or complete bile, components of biliary fluid and pancreatic enzyme secretion, to mimic the duodenum compartment, at 37°C overnight. However, minimum hatching was observed in all conditions after 24 hours (**S3 Fig**). These results suggest that AMCase, at physiologic gastric pH of 2, is critical for *Ascaris* egg hatching *in vitro*.

To verify the role of AMCase and gastric acid in *A. suum* infection *in vivo*, we first assessed production of AMCase and gastric acid production post *Ascaris* infection. Mice were infected

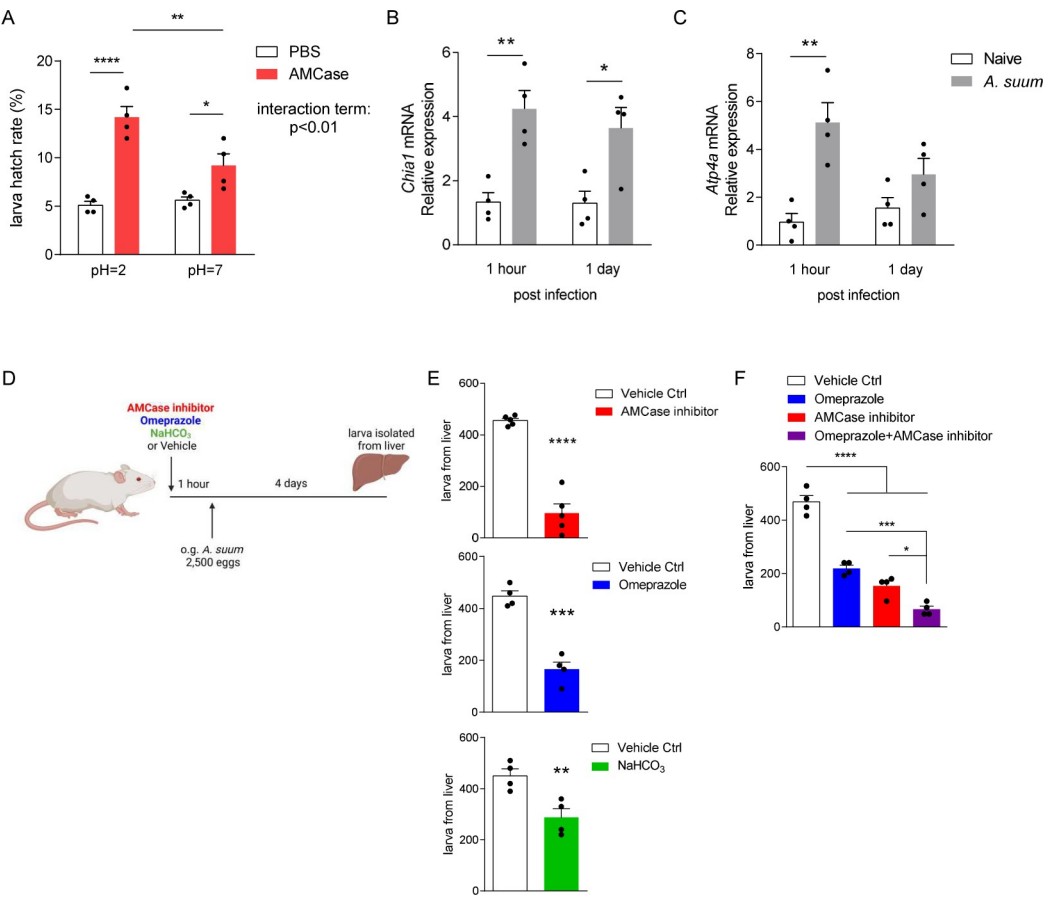

**Fig 2. AMCase in acidic conditions promotes *Ascaris* larval hatching: (A)** *Ascaris* eggs were treated with recombinant AMCase under acidic and neutral pH *in vitro*. Larval hatch rate was quantified post treatment. **(B-C)** BALB/c mice were challenged by oral gavage with 2,500 eggs of *Ascaris* and stomach tissue was harvested 1 hour or 1 day post infection to extract total RNA. *Chia1* **(B)** and *Atp4a* **(C)** expression level were quantified by qPCR. **(D)** BALB/c mice were pretreated with AMCase inhibitor, omeprazole, 10% $NaHCO_3$ or vehicle control before challenged by oral gavage with 2,500 eggs of *Ascaris*. **(E-F)** Larva isolated from the liver 4 days post infection under different conditions was quantified. (n$\geq$4, mean±S. E.M, *p<0.05, **p<0.01, ***p<0.001, ****p<0.0001 using two-way ANOVA followed by Tukey's test for multiple comparison (**A**), two-tailed Student's t-test (**B, C, E**) or one-way ANOVA followed by Tukey's test for multiple comparison (**F**). Data are shown as representative of two independent experiments. Illustration generated by Biorender.com).

with 2,500 eggs of *A. suum* by oral gavage and stomach tissue was removed at predetermined time intervals post infection (1 hour and 1 day). AMCase (*Chia1*) and $H^+/K^+$ ATPase (*Atp4a*) expression was measured by qPCR. Ingestion of *A. suum* eggs upregulated both AMCase (**Fig 2B**) and $H^+$-$K^+$-ATPase (**Fig 2C**) production in the stomach, starting at 1 hour post infection. These results suggest that the stomach acutely responds to *A. suum* eggs by upregulating production of AMCase and gastric acid.

To assess the effect of AMCase and/or gastric acid *in vivo*, we pre-treated mice with the AMCase inhibitor 112252, omeprazole (a commercially available proton pump inhibitor), or 10% sodium bicarbonate by oral gavage once, 1 hour prior to infection with 2,500 *Ascaris* eggs and measured larval burden in the liver at 4 days post infection [15] (**Fig 2D**). Following oral ingestion of *Ascaris* eggs and the initiation of the larval migratory cycle, *Ascaris* larvae migrate via the portal vasculature to the liver at day 4 post infection and subsequently migrate via systemic circulatory system to the lungs at day 8 post infection [15]. Thus, the liver is the first organ system, following oral ingestion of *Ascaris* eggs, that can be evaluated to determine

*Ascaris* larval burden. We found that chitinase inhibition, proton pump inhibition and acid neutralization with sodium bicarbonate all independently decreased the number of larvae migrating to the liver (**Fig 2E**). In addition, to assess if there is a cumulative effect of antagonizing both AMCase and gastric acid simultaneously, we treated mice with AMCase inhibitor and omeprazole prior to *Ascaris* infection by oral gavage. We found that combined inhibition of AMCase and gastric acid significantly reduced larval migration to the liver compared to either treatment alone (**Fig 2F**). Corroborating with our *in vitro* data, we have demonstrated that inhibiting AMCase and gastric acid reduces *A. suum* larval migration likely as a result of reduced egg hatching in the stomach. Together, our data strongly support the concept that AMCase and gastric acid facilitate *A. suum* larval hatching and initiation of the larval migratory cycle.

## Tamoxifen treatment interrupts *A. suum* larval migration

To better delineate the role of the gastric corpus cellular function in larval hatching and larval migration, we utilized the high-dose tamoxifen (TMX) treatment model, in which mice were pretreated with TMX (250 mg/kg) daily for three days followed by infection with *A. suum* eggs by oral gavage (**Fig 3A**). TMX treatment in mice induces parietal cell loss and chief cell reprogramming into spasmolytic polypeptide-expression metaplasia (SPEM) cells. With parietal cell death, acid secretion decreases greatly. And when chief cells reprogram to SPEM cells, they lose most digestive enzyme production, thus likely causing greatly decreased gastric expression of *Chia1*, the gene encoding AMCase [28,29]. We first confirmed this observation by demonstrating that TMX treatment nearly abolished gastric AMCase and H$^+$/K$^+$ ATPase expression by qPCR (**Fig 3B and 3C**).

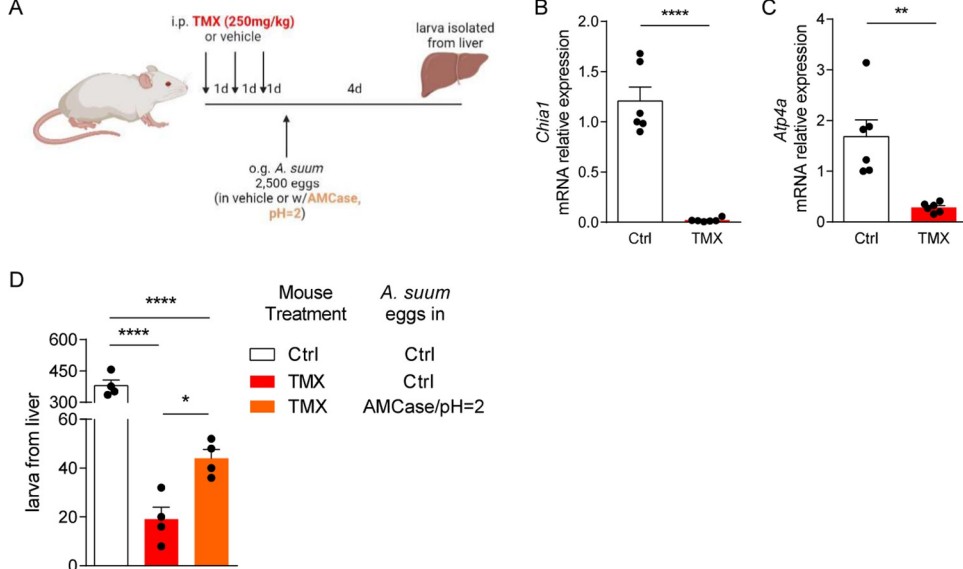

**Fig 3. Tamoxifen reduces AMCase and acid production in stomach which limits *Ascaris* larval migration. (A)** BALB/c mice were pretreated with tamoxifen intraperitoneally before challenged by oral gavage with 2,500 eggs of *Ascaris*, in PBS or vehicle containing recombinant AMCase at pH 2. Reduction of **(B)** AMCase and **(C)** H$^+$-K$^+$ ATPase was confirmed by qPCR. **(D)** Larval isolated from the liver 4 days post infection were quantified. (n≥4, mean±S.E.M, *p<0.05, **p<0.01, ****p<0.0001 using two-tailed Student's t-test **(B, C)** or one-way ANOVA followed by Bonferroni's test for multiple comparison **(D)**. Data are shown as representative of two independent experiments. Illustration generated by Biorender.com).

Next, we infected TMX-treated mice with *A. suum* eggs by oral gavage and measured larval burden in the liver 4 days post infection. Pre-treatment with TMX reduced *Ascaris* larval count in the liver by more than 90% (**Fig 3D**). Simultaneously, we performed a rescue experiment with a subset of TMX-treated mice in which mice were infected with eggs of *A. suum* in vehicle containing recombinant AMCase at pH = 2. In comparison, exogenous treatment of *A. suum* eggs with AMCase in acidic solution permitted increased larval burden in the liver towards baseline levels. This experiment further confirmed that AMCase from chief cells and gastric acid from parietal cells are both critical for the initiation of *A. suum* larval migration *in vivo*.

## Inhibition of AMCase and gastric acid prevents downstream *Ascaris*-induced airway hyperreactivity

Previously, we have reported that *A. suum* larval migration through the lungs induces severe acute airway hyperresponsiveness consistent with allergic airway disease [10]. To assess the effect of AMCase and gastric acid on larval migration and the subsequent development of lung pathology, we pretreated mice with both AMCase inhibitor and omeprazole before infection with 2,500 eggs of *A. suum* as stated above. A subset of mice was euthanized on day 8 post infection to assess larval lung burdens. On day 12 post infection airway reactivity was assessed as previously described[8] (**Fig 4A**). We found significantly reduced larval burdens in the lungs 8 days post infection in mice pretreated with AMCase inhibitor and omeprazole, similar to results from liver (**Fig 4B**). Moreover, we discovered that antagonizing both AMCase and gastric acid production significantly reduced airway hyperresponsiveness as measured by decreased respiratory system resistance ($R_{RS}$) (**Fig 4C**). Inhibition of AMCase and gastric acid prior to *Ascaris* infection also ameliorated *Ascaris* larval migration-induced weight loss (**Fig 4D**), and reduced lung type 2 cytokines (IL-4, IL-5, IL-13), which have been shown to be important mediators of *Ascaris*-induced allergic airway disease[8] (**Fig 4E**). Thus, inhibition of AMCase and gastric acid strongly inhibits *Ascaris* larval migration and reduces pulmonary morbidity in a murine model.

## Discussion

In this study, we have demonstrated for the first time that *A. suum* larvae hatch from ingested eggs in the host stomach and may penetrate the gastric mucosa to initiate the essential larval migratory cycle in a murine model. Larval hatching and penetration of the gastric mucosa occurs as soon as 30 minutes after ingestion of eggs in this model [30]. Mechanistically, we have discovered that *Ascaris* utilizes the unique microenvironment of the gastric corpus to promote efficient hatching across the gastric corpus mucosa.

Chitin is a dietary fiber that provides structure to microbes such as fungi and helminth parasites as well as insects and crustaceans [31]. To support the degradation of chitin, gastric chief cells found in the gastric corpus secrete AMCase in order to hydrolyze chitin [23,24]. Acidic conditions that allow AMCase activity in the gastric corpus are maintained by gastric parietal cell expression of the proton pump $H^+$-$K^+$-ATPase enzyme which modulates cytoplasmic $H^+$ and extracellular $K^+$ exchange [32]. Physiologically, AMCase and gastric acid are responsible for not only breaking down chitinous food and maintaining mucosal homeostasis, but also provide the first line of defense against pathogens containing chitin including both fungi and parasites [23,24,27,33]. Interestingly *Ascaris* egg shells, which are ingested by the host from the external environment, are made of a chitin-protein layer that is 1.5 to 2 μm in thickness. This chitinous egg shell provides protection to the larvae during the developmental phase that occurs within moist, humid soil [34]. Because of this thick chitinase exterior, *Ascaris* eggs can

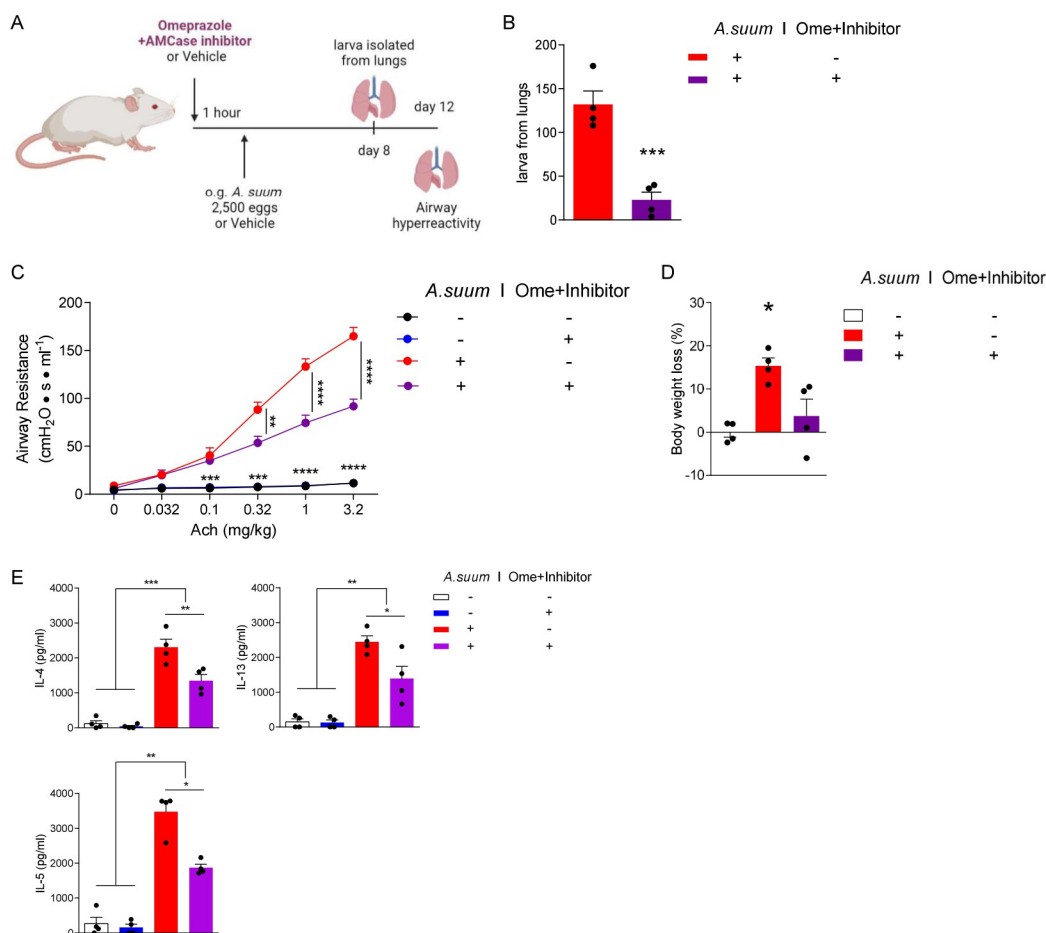

**Fig 4. Omeprazole and AMCase inhibitor reduce airway hyperresponsiveness induced by *Ascaris* larval migration through the lungs** (A) BALB/c mice were pretreated with omeprazole plus AMCase inhibitor or vehicle control before challenge by oral gavage with 2,500 eggs of *Ascaris*. (B) Larval burden was quantified from the lungs 8 days post infection. (C) Respiratory system resistance ($R_{RS}$) was assessed after intravenous injection of increasing doses of acetylcholine (Ach) at 12 days post infection. (D) Body weight loss in mice was measured at 12 days post infection. (E) Type-2 cytokines were quantitated by ELISA from deaggregated lung supernatants. (n = 4, mean±S.E.M, *p<0.05, **p<0.01, ***p<0.001, ****p<0.0001 using two-tailed Student's t-test (**B**) or one-way ANOVA followed by Tukey's test for multiple comparison (**C-E**). Data are shown as representative of two independent experiments. Illustration generated by Biorender.com).

survive for decades in the soil and remain viable [35,36]. However, to cause infection in the host, the chitinous egg shell must be degraded quickly in the host after ingestion. Thus, the gastric corpus microenvironment with abundant acid pumped by parietal cells and acid-dependent chitinase from chief cells, is likely the ideal microenvironment for *Ascaris* egg hatching.

To evaluate the specific role of AMCase and gastric acid in *Ascaris* infection, we specifically antagonized AMCase and gastric acid function using orally administered inhibitors which lead to reduced *Ascaris* larval burden and subsequent reduced lung morbidity. Additionally, when TMX induces acute pyloric metaplasia of the entire gastric corpus with death of parietal cells and conversion of chief cells to a mucous-secreting metaplastic cell lineage, *Ascaris* infectious burden was also reduced. Furthermore, we show that commercially available acid neutralizers and acid inhibitors like omeprazole reduced overall infection and long-term morbidity. Moreover, in addition to the impact on ascariasis, these findings have important implications for other helminths that are transmitted by oral ingestion of eggs, such as

*Trichuris spp*. and *Toxocara spp*. [35,37,38]. It is possible that the gastric microenvironment may have broad application to the infectivity of other parasitic diseases of global significance.

The role of AMCase in helminth immunity has been investigated in other helminths such as *Nippstrongylus brasiliensis*, *Schistosoma mansoni* and *Heligomosomoides polygyrus bakeri* in the lungs during acute infection and in the intestines during chronic infection [39]. Previous studies have shown that AMCase is an essential part of mucosal immunity and induces type 2 immune responses during allergic inflammation in the lungs. Use of an AMCase inhibitor (Bisdionin F) eliminated pulmonary allergic inflammation including eosinophilia in an allergic airway model [40]. Furthermore, AMCase has been shown to influence type 2 immunity in the intestines of *Nippostrongylus* infected mice. Mice deficient in AMCase were found to have reduced type 2 inflammation in the intestines and retention of adult worms suggesting AMCase is critical for the eradication of adult intestinal parasites [39]. Studies evaluating AMCase mediated type 2 immunity in the stomach have yet to be evaluated. However, given the high risk of reinfection, AMCase secretion in the stomach may play a key evolutionary role in parasite eradication during subsequent infection.

Our novel findings are in contradiction to the canonical model of *Ascaris* infection suggesting *Ascaris* larvae hatch and migrate through the intestine in mice [12]. It remains possible that *Ascaris* eggs utilize the gastric corpus microenvironment for hatching but the hatched larvae subsequently translocate across the intestinal mucosa. However, in the *Ascaris* murine model we found no evidence that *Ascaris* larvae initiate infection in the intestines including no evidence of hatched larvae, mucosal damage or inflammatory infiltrate in the duodenum, jejunum, ileum or colon at serial time points post ingestion of *Ascaris* eggs. We did identify hatched larvae along the gastric corpus mucosal membrane and larvae actively translocating the stomach mucosa with development of increased epithelial mucin. Moreover, given the need for the chitinous *Ascaris* eggs to be digested before larval hatching, it makes teleological sense that the anatomical location with the most chitinase in the alimentary tract as well as the low pH needed to activate that chitinase would be where *Ascaris* would hatch. In the current study we used in vitro assays and mouse models to evaluate the gastric corpus microenvironment and did not assay larval hatching in the stomach of pigs or human (the natural hosts of ascariasis). This is an important limitation and focus of our future work as expression of AMCase has been shown to be increased in mice compared to humans [41]. Despite lower concentrations in humans compared to mice, the AMCase and pH microenvironment of the gastric corpus still plays a functional role in human and porcine digestion and immunity. Furthermore, AMCase concentrations in the stomach of pigs has been found to be highly expressed [22]. Thus, our murine model provides the first insight that the gastric corpus microenvironment plays a critical role in *Ascaris* egg hatching and initiation of infection. Additional experiments are being carried out to verify the role of the gastric microenvironment in our porcine *Ascaris* infection model.

*Ascaris* is the most common helminth infection in the world and is a cause of significant life-long morbidity particularly in young children. We have previously shown in a murine model that *Ascaris* larval migration through the host is particularly devastating leading to conditions such as chronic lung disease after a single infection. This study has shown that *Ascaris* larval penetration through the gastric mucosa causes acute gastric damage as well, including reduced gastric vasculature integrity after a single infection, although the long-term consequences remain unknown. However, children living in *Ascaris* endemic regions are constantly re-infected as a result of ingestion of eggs that are ubiquitous within their environment. As a result, repeat damage to the gastric mucosa from recurrent infection may put children at risk for the development of gastric disease such as pyloric metaplasia or potentially predispose them to co-infection with other gastric pathogens like *H. pylori* that share a similar risk factor

profile and geographic niche [42–44]. Conversely, as *H. pylori* causes extensive pyloric meta-plasia in many people [17,45] (as we model here by acute induction with TMX), chronic *H. pylori* infection might eventually be protective against infection with *Ascaris* and other hel-minths that depend on AMCase for hatching. Studies are underway to explore the long-term consequences of *Ascaris* infection on chronic gastric pathology and vice versa.

This study has significant implications for *Ascaris* elimination programs. To date, *Ascaris* disease prevalence has decreased as a result of mass drug administration policies that provide annual or biannual benzimidazole therapy in endemic regions but are unlikely to lead to elimi-nation without additional long-term coordinated interventions [46]. Despite mass drug administration programs, reinfection rates remain high due to the persistence of *Ascaris* eggs within the environment [7]. Furthermore, the current therapeutic interventions available do not prevent infection from occurring but only treat established intestinal ascariasis. Thus, with each reinfection episode, children are at risk of continued larval-induced pathology. Treat-ments that prevent infection are needed and may be the only way to achieve elimination of ascariasis on a global scale. Developing a deeper understanding of the role of the gastric micro-environment in facilitating *Ascaris* infection is the first step to identifying preventative interventions.

## Methods

### Ethics statement

All mice were housed at the American Association for Accreditation of Laboratory Animal Care-accredited vivarium at Baylor College of Medicine under specific-pathogen-free condi-tions. Upon arrival, complete randomization of mice into longitudinal groups was performed. All experimental protocols were approved by the Institutional Animal Care and Use Commit-tee of Baylor College of Medicine and followed federal guidelines (AN-6297).

**Mice.** 8 week-old BALB/c female mice (IMSR_JAX:000651) were purchased from Jackson Laboratories. Female mice were used to ensure consistency in infectious burden [9].

### *A. suum* experimental murine model

Embryonated *A. suum* eggs were provided by Dr. Ricardo Fujiwara, Departamento de Parasi-tologia, Universidade Federal de Minas Gerais, Brazil. BALB/c mice were treated with a single inoculum of 2,500 fully embryonated *A. suum* eggs via oral gavage [15]. The infectious dose of *Ascaris* has been standardized in the literature in order to replicate human disease in a murine model [15,47]. The *A. suum* life cycle in a murine model mimics the life cycle in humans and has been previously described [10,15]. Specifically, *Ascaris* larvae hatch from ingested eggs and travels to the liver, of which the larval burden peaks at day 4 post infection (p.i.). The larvae then travel to the lungs, with maximum larval burden at day 8 p.i., ascend the bronchotracheal tree and are swallowed back into the intestines. Unlike human disease, the *Ascaris* murine model is not competent to sustain chronic adult worms in the intestines. Thus, *Ascaris* larvae are subsequently excreted in the stool by day 14 p.i. The *A. suum* experimental murine model is only capable of evaluating the direct outcomes of *Ascaris* larval migration through the host.

### Collection and preparation of mouse gastrointestinal tissue

After infection of *A. suum*, mice were euthanized at 15 minutes, 30 minutes, 1 hour, 1 day and 4 days p.i. Esophagus, stomachs, small intestines and colon were harvested and fixed in 10% neutral-buffered formalin solution, processed, and embedded in paraffin. 5 $\mu$m sections were cut and slides were stained with Hematoxylin and eosin (H&E) or Periodic Acid Schiff (PAS).

### Gastrointestinal tract vascular permeability

Mice infected with *A. suum* at 1 day p.i. and age-matched, PBS-challenged, naïve mice received 60 mg/kg of Evan's blue dye (E2129, Sigma-Aldrich, St. Louis, MO) injected into the tail vein. The dye circulated for 1 hour and the mice were subsequently euthanized and perfused with saline. Different segments from the gastrointestinal tract, including the stomach, duodenum, jejunum, ileum, cecum and colon were separated and harvested. Tissues were homogenized and centrifuged at 800xg for 10 minutes at room temperature. The supernatant was measured at 620nm in a spectrophotometer and the total extravasated dye in tissue was calculated using a standard curve to determine vascular permeability in the gastrointestinal tract [9,48].

### *In vitro* hatching of *A. suum* eggs

250 eggs of *A. suum* were incubated with or without recombinant acidic mammalian chitinase (AMCase, 9859-GH-050, R&D systems, Minneapolis, MN) at 4 μg/ml, at either pH 2 (RPMI plus HCL) or 7 (RPMI), 37˚C overnight. Larvae were counted afterwards under the microscope. Alternatively, eggs of *A. suum* were incubated at pH = 2 for 30 minutes and then cultured overnight in neutral RPMI (11875093, Thermofisher scientific, Waltham MA) containing 0.25% trypsin (15090046, Thermofisher scientific, Waltham MA), and/or 0.1% bile salt, 2% bovine bile (B8756, B3883, Sigma-Aldrich, St. Louis, MO).

### Isolation of *A. suum* larva from the liver and the lungs *in vivo*

Mice infected with *A. suum* eggs were euthanized 4 days p.i. to assess larval burden in the liver. Liver tissue was harvested and macerated with scissors, suspended into pre-warmed PBS and transferred in a modified Baermann apparatus. The collection system was then filled up to 40 ml of pre-warmed PBS and incubated at 37˚C for 4 hours. Following incubation, the solution containing larvae was collected from the apparatus, centrifuged at 800 xg for 5 minutes at room temperature, and washed with water to remove red blood cells. Larvae were washed with PBS for additional 3 times and counted under the microscope. Alternatively, lung tissue was harvested from euthanized mice 8 days p.i.. Larvae isolation was carried out through the same procedure.

### Inhibition of chitinase and gastric acid in mice

To inhibit gastric chitinase, mice were oral gavaged with human AMCase inhibitor (112252, Sigma-Aldrich, St. Louis, MO) at 5 mg/kg 1 hour prior to infection with *A. suum* eggs. To reduce or neutralize stomach acid, mice were treated with omeprazole (PHR1059, Sigma-Aldrich, St. Louis, MO) at 20 mg/kg i.p.[49] or 100μl 10% $NaHCO_3$[50], 1 hour prior to infection with *A. suum* eggs. Larval burden in the liver was assessed 4 days post infection as described above. Inhibiting both AMCase and gastric acid was done through combining the AMCase inhibitor and omeprazole as stated above at the same time 1 hour before infection.

### Quantitative PCR

Relative expression of mRNA for *Chia1* and *Atp4a* to 18s was detected by two-step, real-time quantitative reverse transcription-polymerase chain reaction (RT-PCR) with the 7500 Real-Time PCR System (Applied Biosystems, Foster City, CA) using Taqman probe (Mm00458221, Mm00444417, Hs03003631 Invitrogen, Carlsbad, CA)[24,41].

### Tamoxifen treatment

Prior to infection with *A. suum* eggs, mice were treated with 250 mg/kg of Tamoxifen (TMX) in corn oil (T5648, Sigma-Aldrich, St. Louis, MO) every day i.p. for 3 days [28,29]. Reduced

AMCase and stomach acid production in the stomach of mice post treatment was confirmed using qPCR (Mm00458221_m1, Mm00444417_m1, Thermofisher scientific, Waltham MA) 24 hours after the last dose. TMX or vehicle treated mice were then infected with 2,500 *A. suum* eggs in PBS. Alternatively, TMX treated mice were infected with 2,500 *A. suum* eggs in PBS containing 4 μg/ml recombinant AMCase in PBS at pH = 2 (adjusted by adding hydrochloric acid to neutral PBS). Larval burden in the liver was assessed at 4 days post infection as described above.

**A. suum induced allergic airway disease.** Mice were treated with or without the combination of AMCase inhibitor and omeprazole 1 hour before infection with 2,500 *A. suum* eggs as described above. At day 12 p.i., body weight loss was quantified. Airway hyperreactivity was determined by measuring respiratory system resistance ($R_{RS}$) in response to increasing dose of acetylcholine (Ach) injected intravenously as previously described [9,10]. ELISA analysis of whole lung for type 2 cytokines IL-4, IL-5, and IL-13 was assessed as previously described (DY 404, DY405, DY413, R&D systems, Minneapolis, MN)[51,52].

## Statistical analysis

Data are presented as means ± standard errors of the means. Significant differences relative to PBS-challenged mice or appropriate controls are expressed by P values of <0.05, as measured two tailed Student's t-test, one-way or two-way ANOVA followed by Tukey's test or Bonferroni's test for multiple comparison. Data normality was confirmed using the Shapiro-Wilk test. Experiments were completed in duplicate.

## Supporting information

**S1 Fig. Intestine of mice post *Ascaris* larva infection.** Wildtype mice were oral gavaged with 2,500 *Ascaris* eggs and euthanized at 12 hours, 24 hours or 4 days post infection. H&E staining were carried out on sections of different intestine tissues.
(PDF)

**S2 Fig. Pepsin does not induce Ascaris larva hatching.** *Ascaris* eggs are treated with 4 or 8 μg/ml of pepsin in pH = 2 overnight. Larvae hatched from the eggs were counted and hatch rate was calculated.
(PDF)

**S3 Fig. *Ascaris* larva does not hatch in intestinal conditions.** *Ascaris* eggs were treated in pH = 2 for 30 minutes and then in different intestinal conditions overnight. Larval hatch rate was then calculated.
(PDF)

## Acknowledgments

The author thanked Dr. David Corry reviewed and edited the manuscript.

## Author Contributions

**Conceptualization:** Yifan Wu, Ricardo Fujiwara, Maria Elena Bottazzi, Jason C. Mills, Jill E. Weatherhead.

**Formal analysis:** Yifan Wu, Jill E. Weatherhead.

**Investigation:** Yifan Wu, Grace Adeniyi-Ipadeola, Mahliyah Adkins-Threats, Matthew Seasock, Charlie Suarez-Reyes, Lizhen Song, Jill E. Weatherhead.

**Supervision:** Yifan Wu, Jill E. Weatherhead.

**Writing – original draft:** Yifan Wu, Jill E. Weatherhead.

**Writing – review & editing:** Yifan Wu, Grace Adeniyi-Ipadeola, Mahliyah Adkins-Threats, Matthew Seasock, Charlie Suarez-Reyes, Ricardo Fujiwara, Maria Elena Bottazzi, Lizhen Song, Jason C. Mills, Jill E. Weatherhead.

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
