## [Decision Letter · Decision Letter 0]

17 Dec 2023

Dear Dr. Weatherhead,

Thank you very much for submitting your manuscript "Host Gastric Corpus Microenvironment Facilitates Ascaris Suum Larval Hatching And Infection in a Murine Model" for consideration at PLOS Neglected Tropical Diseases. As with all papers reviewed by the journal, your manuscript was reviewed by members of the editorial board and by several independent reviewers. The reviewers appreciated the attention to an important topic. Based on the reviews, we are likely to accept this manuscript for publication, providing that you modify the manuscript according to the review recommendations. 

Sincerely,

Aaron R. Jex

Section Editor

Aaron Jex

Section Editor

Reviewer's Responses to Questions

**Key Review Criteria Required for Acceptance?**

**Methods**

-Are the objectives of the study clearly articulated with a clear testable hypothesis stated?

-Is the study design appropriate to address the stated objectives?

-Is the population clearly described and appropriate for the hypothesis being tested?

-Is the sample size sufficient to ensure adequate power to address the hypothesis being tested?

-Were correct statistical analysis used to support conclusions?

-Are there concerns about ethical or regulatory requirements being met?

Reviewer #1: (No Response)

**Results**

-Does the analysis presented match the analysis plan?

-Are the results clearly and completely presented?

-Are the figures (Tables, Images) of sufficient quality for clarity?

Reviewer #1: (No Response)

**Conclusions**

-Are the conclusions supported by the data presented?

-Are the limitations of analysis clearly described?

-Do the authors discuss how these data can be helpful to advance our understanding of the topic under study?

-Is public health relevance addressed?

Reviewer #1: (No Response)

**Editorial and Data Presentation Modifications?**

Reviewer #1: (No Response)

**Summary and General Comments**

Reviewer #1: This is an interesting work providing a comprehensive study of Ascaris larval hatching in a murine model. Overall, this is a well-written manuscript. The experimental design has been conducted in a logically manner. The results are robust, and the discussion of the contradiction finding with previous canonical model is appropriate. Such study will clearly add significantly to the existing scientific knowledge of Ascaris biology. I have a few minor suggestions for authors to consider: 

Please add page and line numbers in the manuscript.

There appears to be errors in citation in the Result section. For example, “Previously, we have reported … with allergic airway disease.” 

1.5 to 2 um” μm

“The role of AMCase in helminth immunity … during chronic infection.” References are needed.

PLOS authors have the option to publish the peer review history of their article (what does this mean?). If published, this will include your full peer review and any attached files.

Reviewer #1: Yes: Tao Wang

Figure Files:

Data Requirements:

Reproducibility:

References

---

## [Editor Report · Decision Letter 1]

21 Jan 2024

Dear Dr. Weatherhead,

We are pleased to inform you that your manuscript 'Host Gastric Corpus Microenvironment Facilitates Ascaris Suum Larval Hatching And Infection in a Murine Model' has been provisionally accepted for publication in PLOS Neglected Tropical Diseases.

Best regards,

Aaron R. Jex

Section Editor

Aaron Jex

Section Editor

---

## [Editor Report · Acceptance letter]

2 Feb 2024

Dear Dr. Weatherhead,

We are delighted to inform you that your manuscript, "Host Gastric Corpus Microenvironment Facilitates Ascaris Suum Larval Hatching And Infection in a Murine Model," has been formally accepted for publication in PLOS Neglected Tropical Diseases.

Best regards,

Shaden Kamhawi

co-Editor-in-Chief

Paul Brindley

co-Editor-in-Chief
